# High-resolution assessment of multidimensional cellular mechanics using label-free refractive-index traction force microscopy

Moosung Lee[1,2,9,10], Hyuntae Jeong[3,10], Chaeyeon Lee[4,10], Mahn Jae Lee[2,5], Benedict Reve Delmo [6], Won Do Heo [4,7✉], Jennifer H. Shin [3✉] & YongKeun Park [1,2,8✉]

A critical requirement for studying cell mechanics is three-dimensional assessment of cellular shapes and forces with high spatiotemporal resolution. Traction force microscopy with fluorescence imaging enables the measurement of cellular forces, but it is limited by photobleaching and a slow acquisition speed. Here, we present refractive-index traction force microscopy (RI-TFM), which simultaneously quantifies the volumetric morphology and traction force of cells using a high-speed illumination scheme with 0.5-Hz temporal resolution. Without labelling, our method enables quantitative analyses of dry-mass distributions and shear (in-plane) and normal (out-of-plane) tractions of single cells on the extracellular matrix. When combined with a constrained total variation-based deconvolution algorithm, it provides 0.55-Pa shear and 1.59-Pa normal traction sensitivity for a 1-kPa hydrogel substrate. We demonstrate its utility by assessing the effects of compromised intracellular stress and capturing the rapid dynamics of cellular junction formation in the spatiotemporal changes in non-planar traction components.

[1] Department of Physics, Korea Advanced Institute of Science and Technology (KAIST), Daejeon 34141, South Korea. [2] KAIST Institute for Health Science and Technology, KAIST, Daejeon 34141, South Korea. [3] Department of Mechanical Engineering, Korea Advanced Institute of Science and Technology (KAIST), Daejeon 34141, South Korea. [4] Department of Biological Sciences, Korea Advanced Institute of Science and Technology (KAIST), Daejeon 34141, South Korea. [5] Graduate School of Medical Science and Engineering, Korea Advanced Institute of Science and Technology (KAIST), Daejeon 34141, South Korea. [6] Department of Bio and Brain Engineering, Korea Advanced Institute of Science and Technology (KAIST), Daejeon 34141, South Korea. [7] KAIST Institute for the BioCentury (KIB), KAIST, Jaejeo, Daejeon 34141, South Korea. [8] Tomocube Inc., Daejeon 34109, South Korea. [9] Present address: Institute for Functional Matter and Quantum Technologies, Universität Stuttgart, 70569 Stuttgart, Germany. [10] These authors contributed equally: Moosung Lee, Hyuntae Jeong, Chaeyeon Lee. ✉email: wondo@kaist.ac.kr; j_shin@kaist.ac.kr; yk.park@kaist.ac.kr

Living cells interact mechanically with their surrounding microenvironment during a wide range of physiological functions, including migration, differentiation, morphogenesis, and immune responses[1,2]. Understanding the mechanoresponses can reveal the correlations between these cellular functions and the cellular mechanics that govern them. This requires simultaneous access to cellular forces and structures with high spatiotemporal resolution. Traction force microscopy (TFM) combined with fluorescence microscopy (FL) has traditionally been used to evaluate cellular forces[3,4]. TFM measures cell-generated forces by imaging the displacement of fluorescent markers in a deformed elastomer[5,6]. However, the utilization of FL-TFM is generally limited by its slow acquisition speed, photobleaching, and phototoxicity. These limitations impede long-term, high-speed, and three-dimensional (3D) analyses of cellular mechanics.

To circumvent the challenges in FL-TFM, label-free TFM methods have recently been proposed. Elastic resonator interference stress microscopy, for example, uses an elastic optical micro-resonator as a substrate and converts the interference intensity of reflected light into cell-induced stress[7]. However, this technique does not resolve the 3D direction of cellular traction. Label-free TFM in biopolymer networks has also been demonstrated using optical coherence tomography, albeit with limited sensitivity and resolution[8]. Alternatively, quantitative phase imaging (QPI) techniques[9–11] are promising for label-free TFM with sub-micrometer spatial resolution. By exploiting refractive index (RI) as an intrinsic image contrast, QPI techniques enable quantification of dry cell mass, rapid cell dynamics, and identification of unlabelled microparticles at high resolution[9]. Previously, 2D-QPI has been utilized in TFM, which had limited accuracy and difficulty in measuring axial traction[12]. To overcome this, 3D-QPI techniques are essential for general applications in cellular mechanics.

Here, we present refractive index traction force microscopy (RI-TFM), a label-free 3D and high-speed TFM by exploiting RI tomography. By implementing a high-speed angular-scanning scheme, RI-TFM simultaneously captures the 3D dry-mass distribution and traction stresses of living cells mounted on a planar hydrogel. With a 0.5-Hz maximal volumetric acquisition rate, we studied the dynamic changes in the spatiotemporal distribution of proteins and cellular tractions when three different types of cells, namely NIH-3T3, MDCK, and CD8[+] T cells, were challenged by disruptive chemical treatment, low temperature, and fortified cellular junctions, respectively.

## Results

### Principle of RI-TFM.
Figure 1 is a schematic of the RI-TFM, which is described in detail in the Methods. Our optical setup allowed high-speed angular scanning of an incident monochromatic plane wave using a digital micromirror device[13] (Fig. 1a and Supplementary Fig. 1). We imaged a live cell attached to a polyacrylamide substrate with a thickness ranging between 18 and 40 μm. As fiducial markers to estimate the 3D map of a gel deformation vector, $\mathbf{u}(\mathbf{r})$, 200-nm-diameter polystyrene beads were embedded in the substrate with the number density of three per μm³. The stiffness of the gel was set to 11 and 1.2 kPa for two adherent cell types (NIH-3T3 and MDCK) and non-adherent T cells[14], respectively (see Methods and Supplementary Fig. 2). The transmitted field images of the sample were holographically recorded with 91 different incident angles of plane waves using an off-axis Mach–Zehnder interferometer. From the obtained holograms, we reconstructed a 3D RI tomogram using a constrained total variation deconvolution algorithm[15,16]. The spatial resolution of the tomogram was previously measured to be 161 nm

lateral and 401 nm axial[17]. In our study, the reconstructed tomogram was down-sampled twice to accelerate the reconstruction speed while maintaining the distinctness of the fiducial markers (Supplementary Fig. 3).

In the representative RI tomogram of NIH-3T3 fibroblast cell expressing paxillin-EGFP, the contact between the cell and the planar hydrogel substrate was clearly visualized (see Methods) (Fig. 1b). For comparison with FL-TFM, we utilized 3D structured illumination microscopy (3D-SIM) to obtain high-resolution FL images[18]. Both FL and RI images consistently distinguished the microbeads in the substrate from the attached fibroblast. We also compared the 3D deformation maps of both imaging modalities by estimating the distribution of 3D displacement (Fig. 1c; see Methods). For RI-TFM, we manually removed the cellular region, applied a high-pass filter, and estimated the 3D displacement of beads from the post-processed tomogram (see Methods). The 3D displacement of beads was clearly visible in an overlay of the bead positions before and after applying traction stress (in magenta and cyan, respectively). Utilizing a rapid Fourier transform-based image correlation approach, we quantified the 3D displacement maps of the substrate[4,19] (Fig. 1d). The typical subset volume and grid spacing were set as $64^3$ voxels ($11.7^3$ μm³) and 11 voxels (2 μm), respectively. The root-mean-square errors (RMSEs) between these methods were 4.89, 10.81, and 11.36 nm along the $x$, $y$, and $z$ directions, respectively. Considering that these values are far lower than the imaging resolution, we conclude that the result validated the consistency of both methods.

From the 3D displacement map, $\mathbf{u}(\mathbf{r})$, we retrieved the multidimensional traction map, $\mathbf{T}(x, y; z = 0)$, composed of shear ($T_x$ and $T_y$; in-plane) and normal ($T_z$; out-of-plane) tractions (Fig. 1e). Solving this problem is generally known to be an ill-posed problem due to the limited measurement range of the displacement map[20]. To resolve this issue, we implemented a constrained total variation deconvolution algorithm, which is similar to our RI reconstruction algorithm (see Methods). Consistently with a previous study, our results from RI-TFM showed that the inwardly oriented shear tractions reaching 492.5 Pa significantly appeared in the vicinity of the focal adhesions[3]. Also, we observed the change in the direction of the normal traction vector at the cell-substrate anchorages, which was 477.4 Pa upward and 270.9 Pa downward at the distal (towards the cell periphery) and proximal (towards the cell center) parts of the cell ends, respectively.

Although the results of RI-TFM and FL-TFM are in good agreement, the latter exhibited pointwise artifacts. We attribute these artifacts to the long acquisition time of the technique, which exceeded 10 min and made the technique susceptible to mechanical noises and motion artifacts. 3D FL-SIM requires at least 30 acquisitions per z stack (3 illumination angles times 5 phase shifts times 2 planes/resolved thickness) to achieve resolution doubling[21]. These multiple acquisitions significantly reduce temporal resolution for 3D imaging. In addition, the significant photobleaching and the vignetting problems in SIM also restricted the consistent reconstruction performance across our entire field of view despite a proper post-correction process[22] (Supplementary Fig. 4). In contrast, RI-TFM is label-free and acquires a single tomogram within 1.5 s which is only limited by the camera acquisition rate. Consequently, RI-TFM captures the intricate 3D mechanical and morphological information of living cells with better fidelity than traditional FL-TFM methods, enabling long-term and high-speed 3D assessment.

### Precision and accuracy of RI-TFM.
To compare the performance between RI-TFM and FL-TFM, we quantified the statistics

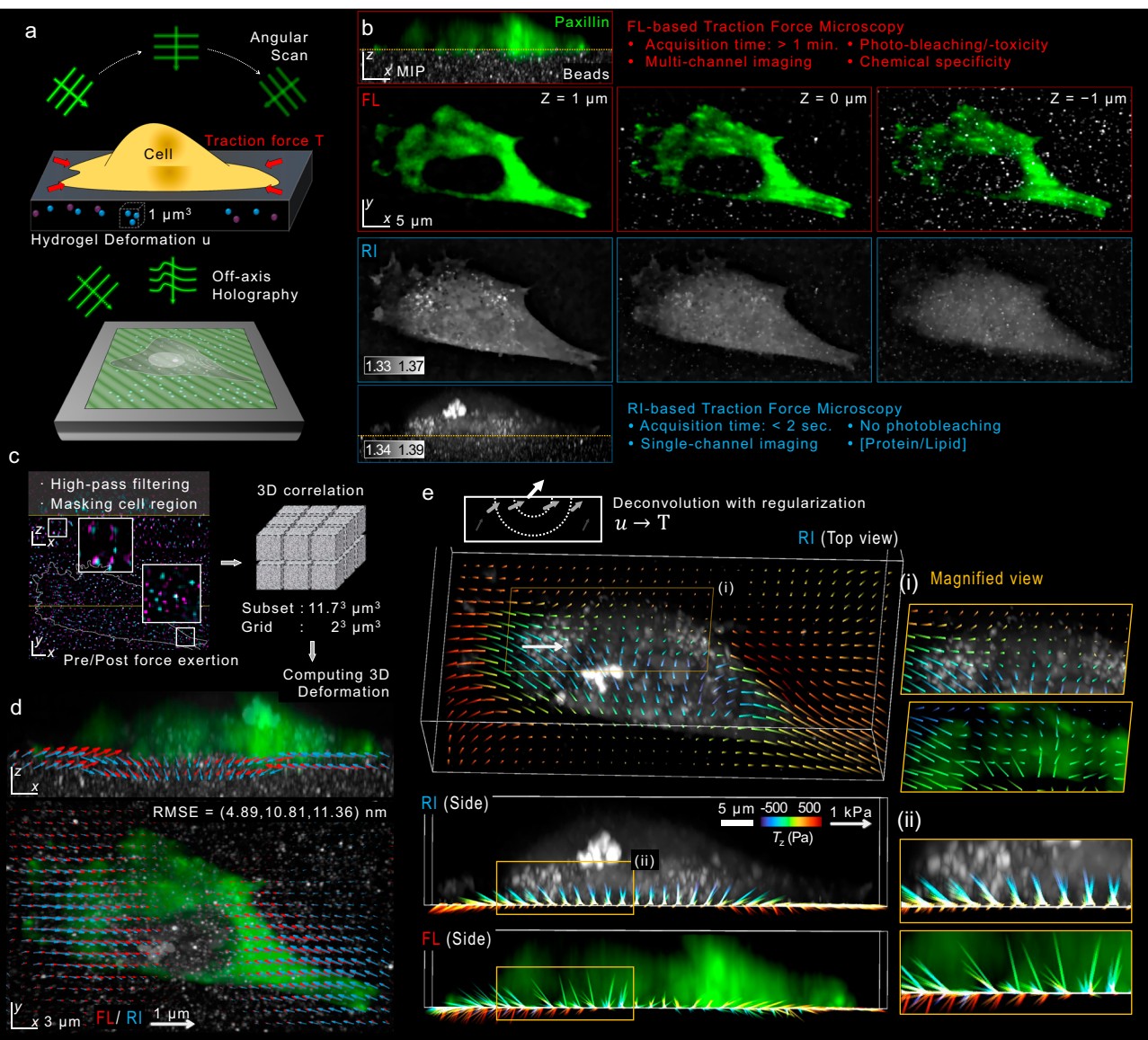

**Fig. 1 Layout of refractive-index traction force microscopy (RI-TFM). a** Schematic. We implemented RI tomography using an off-axis holographic microscope to image a live cell on an elastic polyacrylamide hydrogel embedding 200-nm-diameter polystyrene microspheres. The schematic diagram of the optical setup is illustrated in Supplementary Fig. 1. **b** XZ- and XY-sliced FL and RI images of a fibroblast cell expressing paxillin-EGFP, which was mounted on an 11-kPa hydrogel. **c** Postprocessing. The hydrogel was imaged twice: before (magenta) and after (cyan), the cell was chemically detached using an EDTA solution. The acquired tomograms were high-pass filtered, masked, and registered. Inset images show magnified regions where 3D displacements of the embedded beads were significant. **d** 3D displacement vectors were computed using a correlation method based on the Fourier transform and drawn in the maximum RI projections of the cell. RMSE root-mean-squared error. **e** Estimated multidimensional traction of the cell using RI-TFM was compared with that of FL-TFM.

of the multidimensional traction stress of the two methods (Fig. 2). Because FL-TFM with 3D-SIM was susceptible to reconstruction artifacts, we used a conventional widefield illumination scheme in this experiment. To estimate the accuracy and precision of the traction measurement, we translated a hydrogel substrate using a motorized XY stage (MLS203-1; Thorlabs) by 0.5–3 μm with an interval of 500 nm, imaged the tomograms and measured 3D displacement maps. Ideally, an estimated displacement map should estimate the stage displacement with zero standard deviations. Representative images show that both imaging modalities accurately measured the displacement of the substrate with sub-voxel precision (Fig. 2a). For RI-TFM, we questioned whether the signals from the mounted samples on the substrate might deteriorate the measurement

performance of RI-TFM. We tested this by dispersing silica microspheres having 2-μm-diameter on the hydrogel and performing a similar experiment. As a result, all of the results provided similar lateral precision, with <10-nm standard deviations (Fig. 2b). The axial precision of RI-TFM, however, outperformed widefield FL microscopy by 3.6 times.

We additionally estimated the measurement accuracy by computing the RMSEs from the predicted displacement, which was twice the accuracy of RI-TFM. Time-lapse measurements for 1120 s also showed temporally robust accuracy and precision, with 3D RMSEs of 6.12, 5.92, and 14.89 nm along the $x$, $y$, and $z$ directions, respectively (Supplementary Video 1).

The reconstructed traction force per unit area further revealed superior precision of RI-TFM (Fig. 2c). This method

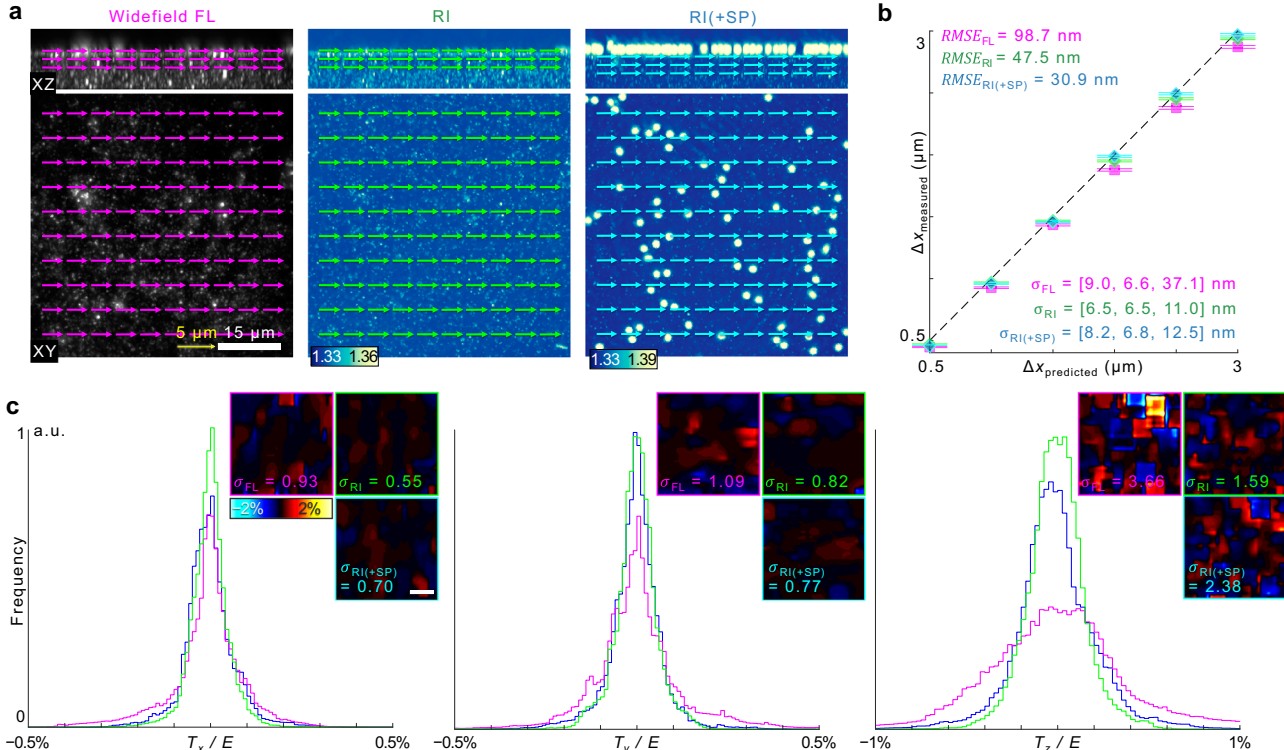

**Fig. 2 Accuracy and precision of RI-TFM. a** Representative fluorescence and RI tomograms. 3D displacement vectors were estimated when the hydrogel was laterally displaced by 3 μm. Arrows denote the estimated displacement vectors at each spacing grid. **b** Statistics of measured displacement vectors. Points and horizontal lines indicate the mean and mean ± standard deviation, respectively. The number of data points per statistic is >2700. The dashed line represents the expected linear plot. **c** Measurement sensitivity of multidimensional traction stress per Young's modulus was defined as its standard deviation. σ standard deviation, RMSE root mean squared error.

outperformed FL-TFM and provided maximum 0.55-Pa lateral and 1.59-Pa axial stress sensitivities for a 1-kPa hydrogel substrate. These combined findings confirmed the feasibility of using RI-TFM to monitor traction forces with accuracy and precision throughout the experiment.

**RI-TFM captures the effect of chemical cytoskeleton disruption on cellular traction force.** To leverage our method's capability of precisely measuring the cellular traction and morphologies, we investigated the effect of compromised actin networks on 3D cellular morphology and traction force. To disrupt F-actin organization, we treated NIH-3T3 fibroblasts with 5 μM cytochalasin D and imaged them after 1 h (Fig. 3a). Both FL and RI images indicated that disruption of actin caused weakening of focal adhesions, leading the distal ends to shrink and round up (Fig. 3a). Consequently, the overall traction at the cell-substrate interface significantly decreased, and residual normal traction appeared at the proximal adhesions[23]. Before cytochalasin D treatment, the maximal shear/normal tractions of the cells appeared in the vicinity of cell-substrate anchorages and reached 620.2/1223.2 Pa for cell 1 and 1510.6/681.7 Pa for cell 2, respectively. Consistent with our observation in Fig. 1, the direction of the normal traction (upward) at the distal ends was opposite to that at the proximal ends (downward). These general features indicate that the cell-substrate adhesions anchored at the distal cell ends were pulled towards the center, creating torque on the extracellular matrix[24]. After the chemical inhibition of the F-actin polymerization, the maximum traction stresses at these cell-substrate adhesions decreased significantly to 130.0/356.9 Pa for cell 1 and 168.8/241.4 Pa for cell 2, respectively.

We quantified the statistics of these observations using histograms of the cell-mass parameters and multidimensional

traction stresses (Fig. 3b). We first assessed the 3D morphological parameters of the cells. Because the relative RI of the cell is proportional to the cell dry-mass[25], we could quantitatively estimate the cell dry-mass densities from the reconstructed 3D tomogram of the cells. Because we wanted to test whether the gravitational pressure by the cell mass might affect the estimated normal tractions, we converted a 2D projected dry-mass density map into the map of pressure by cell mass (see Methods). The results indicated that the pressure by cell weight was a few mPa, which was five orders of magnitude smaller than the estimated shear and normal traction forces. Consistent with our observations, the average pressure owing to dry-mass densities of cells decreased by 10%, and the standard deviation increased. Additionally, the mean ± standard deviations of the shear/normal tractions were (117.2 ± 102.3)/(113.0 ± 94.6) Pa for cell 1 and (287.3 ± 141.4)/(186.0 ± 112.7) Pa for cell 2, respectively. After cytochalasin D was treated, the average shear tractions decreased by 5.1 and 2.7 times for cell 1 and cell 2, respectively. Such drastic attenuation of the shear traction is consistent with a previous study using magnetic twisting cytometry[26]. Correspondingly, we observed that the normal tractions also decreased by 4.3 and 1.9 times for cell 1 and 2, respectively. As a result, an adherent cell's overall multidimensional cell traction force is predominantly produced by a dynamic actin network.

**RI-TFM captures the rapid surface contraction during the cell rear-detachment mode.** The high-speed imaging capability of RI-TFM is useful for capturing sudden morphological changes in living cells. We demonstrated this by imaging the rapid contraction of a Madin-Darby canine kidney (MDCK) cell during its

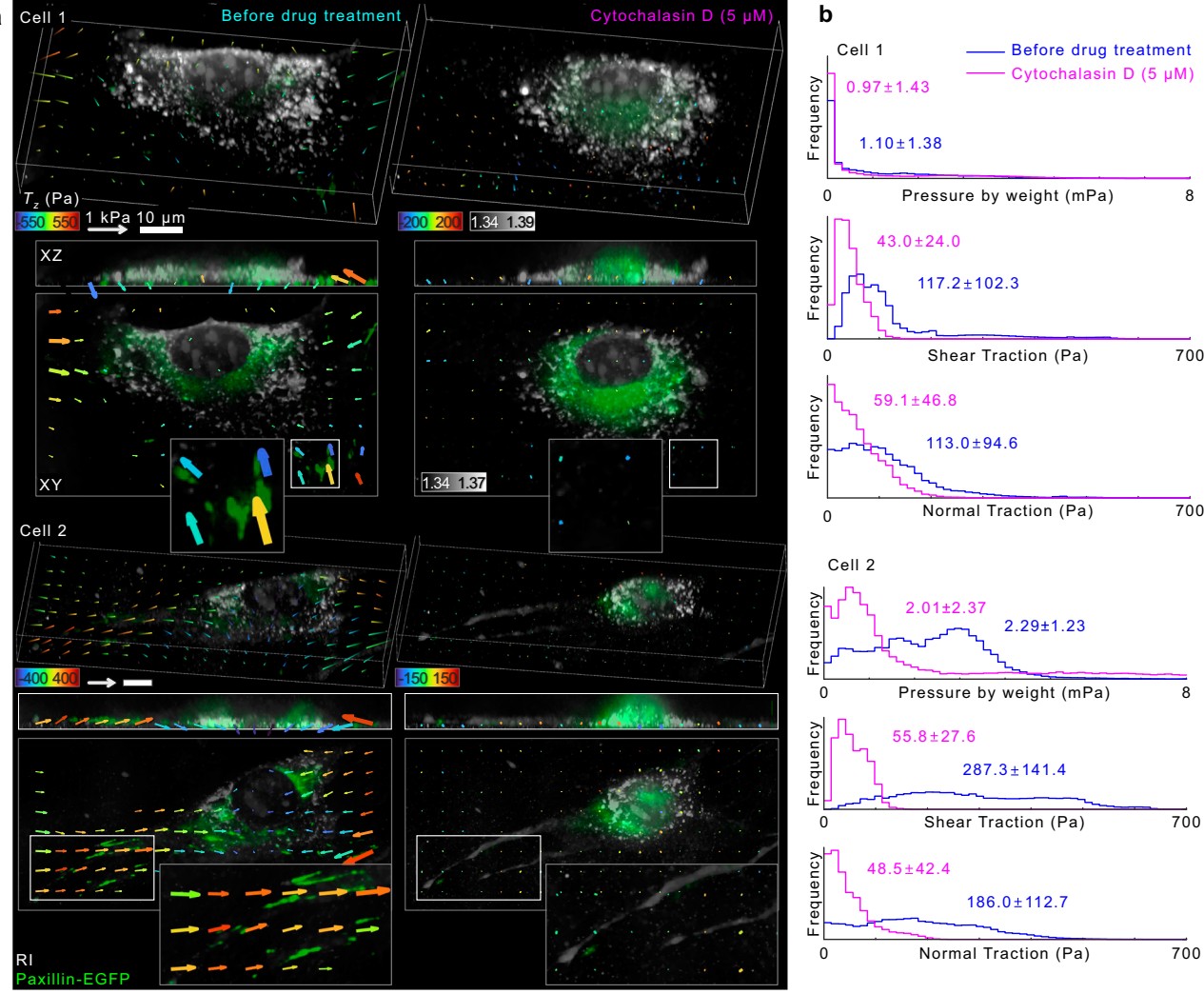

**Fig. 3 3D RI and traction of NIH-3T3 fibroblasts expressing paxillin-EGFP before and after treatment of cytochalasin D. a** Representative images of overlaid 3D-rendered tomograms of RI (grey) and FL (paxillin-EGFP; green) and traction stress for NIH3T3 cell 1 (top) and 2 (below) before and 1 h after treatment with cytochalasin D (5 μM). For clearer visualization, we down-sampled the traction vectors and drew significant traction vectors above threshold values in the XY images. **b** Histograms of pressure by weight, shear traction, and normal traction before (magenta) and after (cyan) cytochalasin D treatment. The mean ± standard deviation for each histogram is presented.

rear-detachment mode (Fig. 4). We disrupted the viable condition of the MDCK cell by lowering the medium temperature to room temperature (25 °C) and observed the dynamics of the cell. RI-TFM resolved the interface between the MDCK cell and the substrate and the 3D displacement of the beads due to cellular traction (Supplementary Video 2). We acquired 23 sequences of the 3D tomograms and the traction stresses for 880 s with intervals of 40 s (Supplementary Video 3). We quantified the multidimensional tractions, which showed that the cell maintained the cellular traction in the vicinity of cell edges before the rear detachment (Fig. 4a). In the initial stage (40 s), the magnitude of inwardly oriented shear traction generated by the cell reached 330 Pa[27] (Fig. 4b). We observed significant normal traction at the three cell edges, reaching a magnitude of 180 Pa. Similar to the results of NIH-3T3 cells, the directions of normal tractions were upward at the distal (towards the cell periphery) and downward at the proximal (towards the cell center) parts of the cell ends. These observations imply the colocalization of focal adhesions in the cell edges.

As shown in Fig. 4a, after 480 s, we observed that the left cell edge started contracting toward the cell body. Interestingly, we

identified the increasing density of the surface dry-mass distribution in the vicinity of the contracting cell edge (Fig. 4b; 560 s). The surface dry-mass density was maximized at 600 s when the rear detachment occurred. At this moment, the XZ intensity maximum image of the 3D RI tomogram clarified that the left cell edge was completely detached. The temporal change in the cell traction, $\Delta \mathbf{T}_j = \mathbf{T}_j - \mathbf{T}_{j-1}$, indicated that the deformed substrate at the distal end was restored while the attached cell rear still pulled the substrate with the upward normal traction reaching 119.9 Pa. Both the increased surface dry mass and the traction stress in the vicinity of the detached cell rear were recovered to steady states until 800 s.

To further visualize and quantify the instantaneous changes at 600 s, we used kymographs of the 3D morphology and traction stress (Fig. 4c, d). At 600 s, the average surface dry-mass density reached the maximum of 20.45 fg/μm² at 4.8 μm from the distal end (Fig. 4c). The maximal changes in the shear and normal tractions were −169.1 (opposite to the cell body) and 139.3 Pa (upward), respectively (Fig. 4d). The collective results suggest that RI-TFM provides insight into the rapid changes of 3D cellular structures and mechanics.

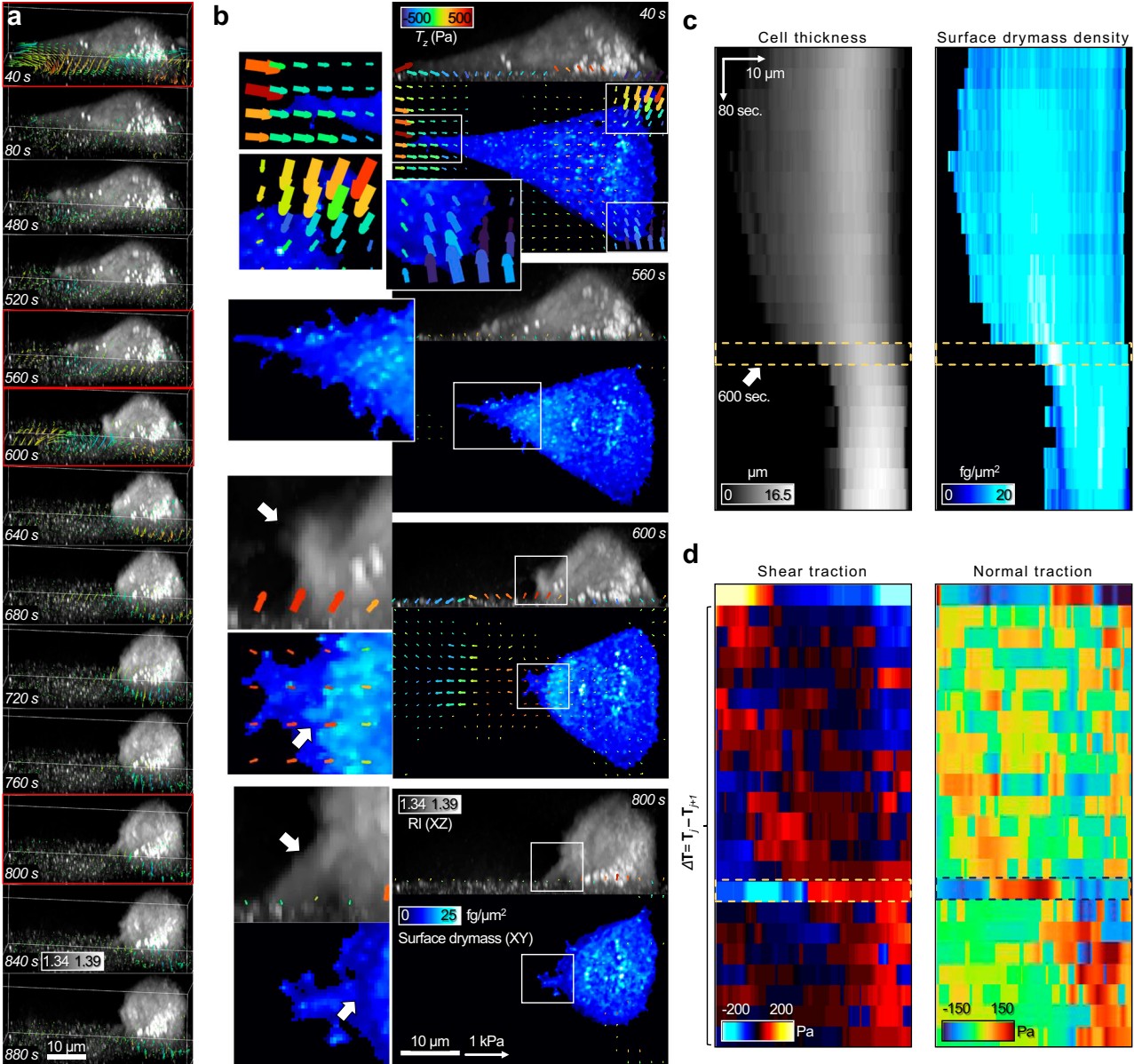

**Fig. 4 Sequential images of the MDCK cell detached from an 11-kPa gel substrate. a** 3D-rendered time sequences. Vectors indicate the differential traction stress, $\Delta \mathbf{T}_j = \mathbf{T}_j - \mathbf{T}_{j-1}$, where $j$ is the frame index. Here, we defined $\mathbf{T}_{0\ s} = \mathbf{0}$. **b** XZ maximum intensity projections, XY surface dry-mass distributions, and multidimensional traction stress maps of selected sequences in (**a**) (red). White arrows indicate regions related to the detached focal adhesion. For clearer visualization, significant traction vectors above threshold values were drawn in the XY images. **c**, **d** Kymographs along the $x$ direction for (**c**), 3D morphological parameters (maximum cell thickness and average surface dry-mass density), **d** maximal value for the differential shear and normal traction. See Supplementary Videos 2 and 3 for more details.

**RI-TFM traces out cell mechanics of cytotoxic T cells.** The mechanical force generated by T cells is correlated with their cytotoxicity[28]. In particular, the traction force generated by T cells may change dynamically during the formation of an immunological synapse, which is a nanoscale intercellular junction between a T cell and an antigen-presenting cell. Although extremely challenging in conventional methods, quantitative assessment of their multidimensional relationship is important for potential applications in immune cell biology and immune oncology[29]. This is possible with high-speed RI-TFM using a soft hydrogel substrate coated with CD3, CD28, and T cell-specific antibodies (see Methods). We demonstrated the measurement of multidimensional traction force generated by human CD8[+] cytotoxic T lymphocytes (CTLs) (Fig. 5).

We first compared the 3D dynamics of resting and activated CTLs (Fig. 5a, b; Supplementary Video 4). The 3D RI images showed that the volume of an activated CTL was larger than that of a resting CTL. Time-lapse measurements showed that the resting CTL did not generate a uniform polarity in traction directionality during the observation period (Fig. 5a). In contrast, the activated CTL formed a radially symmetric structure and traction force per unit area at the interface (Fig. 5b). Prior to the formation of a stable synapse at 100 sec, the activated CTL exerted instantaneous downward normal traction on the interface. The average normal traction peaked at $95 \pm 25$ Pa (60 s), which may be associated with the approaching dynamics during the formation of immunological synapse[30] (Fig. 5c). The average normal traction then decreased to 40 Pa and maintained a steady

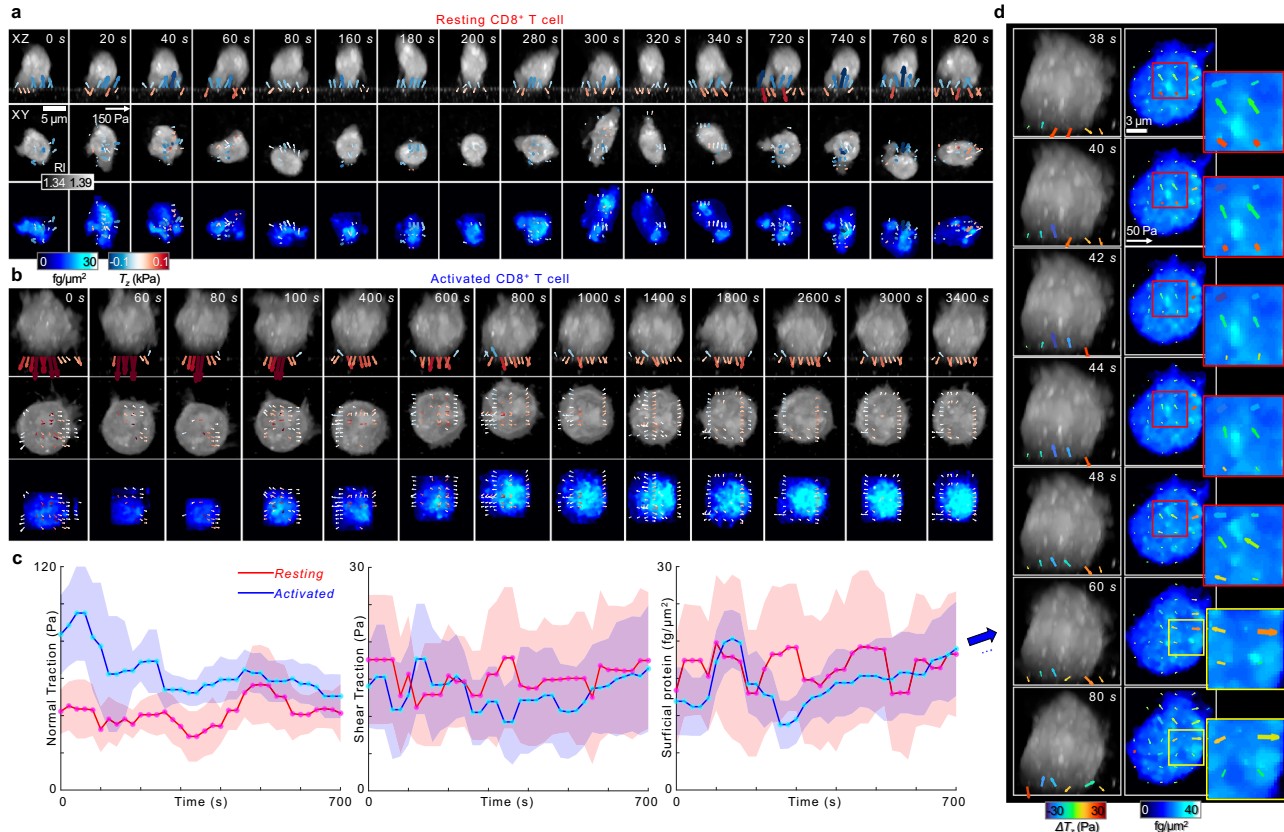

**Fig. 5 Quantification of the multidimensional traction forces generated by human CD8+ cytotoxic T lymphocytes (CTLs). a, b,** Representative sequential snapshots of (**a**) resting and (**b**) activated CTL on a 1.2-kPa gel coated with anti-CD3/CD28 antibodies. For clearer visualization, significant traction vectors above threshold values were drawn in the XY images. **c** Temporal changes in normal traction, shear traction (second row), and surface dry-mass densities for resting and activated CTLs in (**a**, **b**). The central points and shades indicate the mean and standard deviation (number of points > 50), respectively. **d,** Representative sequential snapshots of activated CTLs acquired at an interval of 2 s. See Supplementary Videos 4 and 5 for more details.

state after 200 s. Interestingly, the shear traction correspondingly started to increase gradually from 11.0 ± 6.1 Pa at 200 s. In particular, the directions of the shear traction exhibited the radially symmetric structure, indicating that the CTL started to spread the cell membrane along the substrate. The surface dry-mass distribution also exhibited a radially symmetric structure with a higher concentration at the center than at the periphery. The mean surface density reached the maximum at 3240 s ($16.4 ± 8.4$ fg/$\mu$m$^2$). We have compared the statistics of these parameters for 6 resting and 10 activated CTLs 30 min after an interaction at the substrate, which showed a consistent trend with statistically significant differences (Supplementary Fig. 5).

The structure of the surface protein density reflects radially hierarchical structures of the immunological synapse, which is formed by the engagement of T cell receptors (TCRs) with peptide-loaded MHC complexes[31]. This is in stark contrast to the asymmetric and specular protein distribution at the synapse formed by the chimeric antigen receptor, a genetically re-engineered protein that adopts a single-chain variable fragment of monoclonal antibodies[17,32]. Because the substrate in our study lacked the relevant receptors for the formation of the immunological synapse, the results suggest that such a characteristic structure of the TCR immunological synapse may also be triggered by CD3 and CD28 antibodies.

Finally, we demonstrated a high-speed quantitative analysis of surface dry-mass densities and multidimensional traction stresses of activated CTLs at a 0.5-Hz acquisition rate (Supplementary Video 5). To elucidate the temporal changes of the cellular traction more clearly, we quantified the relative cellular traction,

$$\Delta \mathbf{T}(\mathbf{r}, t) = \mathbf{T}(\mathbf{r}, t) - \mathbf{T}(\mathbf{r}, 0).$$ As a result of the correlative analysis of $\Delta \mathbf{T}$ and the surface dry-mass densities, we observed the parallel directionality of the weak lateral traction and dry-mass-rich condensates (Fig. 5d). As a prominent example, we identified a dry-mass-rich condensate at 38 s whose major-axis diameter and dry-mass density were 1.26 $\mu$m and 30.1 fg/$\mu$m$^2$, respectively. This condensate is located on the left upper part of the cell surface, along with the lateral traction of 13.8 Pa in a parallel direction. During the locomotion, the size and the traction diminished gradually until its disappearance after 60 s. In other regions, such condensates appeared stochastically and moved randomly. These results may be associated with the rapid microvillar search dynamics, which were previously observed using lattice light-sheet microscopy[33]. Taken together, the collective results demonstrate the versatile applicability of RI-TFM for quantitative analysis in which rapid immune cell mechanics should be investigated.

## Discussion

We have shown here that RI tomography can be a unique imaging option for assessing 3D cellular shapes and multidimensional traction forces. Contrary to FL microscopy, which requires a multichannel imaging setup to image cells and gel substrates separately, our method extracts both 3D cellular dry-mass and nonplanar traction force from a 3D RI tomogram. Because the RI can be obtained with a robust high-speed imaging setup without exogenous labeling, our method allows for high-throughput analysis via long-term, high-speed imaging.

This feature of our method provides unique advantages compared with various FL-based TFM methods. For example, compared with the widefield FL methods, our method provides higher spatial resolution, which is essential for improving the measurement precision of TFM (Supplementary Fig. 6). Our method provides the spatial resolution and comparable measurement precision comparable to that of SIM[34], but our method provides a more consistent performance with a faster acquisition speed and more robustness against reconstruction artifacts (Supplementary Fig. 4). In addition, compared with 3D FL TFM using point spread function estimation[35], our method provides the 3D tomogram across 50 μm, which can access whole-3D information about the cell feature and traction.

The data presented here not only recapitulate previous findings from FL-TFM but also reveal interesting phenomena that are hardly accessible without quantifying the 3D protein concentration with high spatiotemporal resolution. In agreement with previous studies, we have elucidated the following general features of cell mechanics: (1) rotational moments are generated by adherent cells at their focal adhesions;[24] (2) downward normal tractions are generated by spherically shaped cells[36], and (3) traction forces decrease by the chemical disruption of F-actin polymerization[26]. Few studies, however, have addressed the long-held questions regarding the correlation between the cellular structures and forces, including (1) how the traction forces are affected during the abrupt shape changes of living cells; (2) how the cell density is organized when cells interact with its surrounding environment; (3) how the multidimensional traction forces correspondingly change during these rapid events. We could challenge these questions because the proposed method could provide high-speed imaging capabilities of both the cell dry mass and traction force at the same time.

The implemented system now provides 0.5-Hz temporal resolution, 0.55-Pa shear (in-plane), and 1.59-Pa normal (out-of-plane) traction sensitivity for a 1-kPa hydrogel substrate. These could be improved in many technical aspects. The camera acquisition speed currently limits the acquisition speed, which can be improved to a few tens of Hz by utilizing a high-speed camera and sparse measurement[37]. We may also outperform the current spatial resolution and traction sensitivity by exploiting advanced regularization algorithms[16,38,39], deep-learning-based segmentation of fiducial markers and cells[17,40,41], and low-noise illumination sources[42]. To achieve the practical advantages in reference-free measurements and studies in a 3D microenvironment, we also anticipate the replacement of substrates[43,44].

A critical limitation of our technique is the lack of chemical specificity, which may limit its range of applications. However, we have shown that we could attain both quantitative analysis and specificity by developing correlative approaches with FL microscopy[17,40,45]. Instead of conventional brightfield methods such as phase contrast microscopy, the proposed method could be a competent alternative to assist FL-based analysis in various areas of cell biology. In particular, RI-TFM could become a unique asset in developmental biology and immune cell biology[46]. In the former applications, our method may provide morphological and mechanical biomarkers that are relevant for predicting the fate of stem cells in native states[47]. In the latter applications, our technique can give access to the rapid 3D mechanics during various immune responses[6,28,48,49] and signalling[50–52]. The utilization of birefringence images can also be used to retrieve cytoskeleton information further and to be systematically analyzed with RI-TFM[53,54]. Ultimately, we envisage the potential use of RI-TFM in immune oncology, where biophysical markers for efficacious immune cell therapy against cancer cells should be revealed[29,32,55].

## Methods

**Optical setup**. We exploited a custom-built ODT system[17]. A plane wave from a continuous green laser (SambaTM 532 nm laser, maximum power = 100 mW, Cobolt) illuminated a sample. The plane wave was separated by a half-wave plate (WPH10M-532, Thorlabs) and the first polarizing beam splitter (PBS251, Thorlabs). The incident plane at the sample path wave was then diffracted by a digital micromirror device (DMD; V-9601, Vialux) for high-speed angular scanning[13] and angularly magnified by a 4-f array of lens 1 ($f = 250$ mm) and a condenser lens (LUMFLN60XW, NA = 1.1, Olympus). An objective lens with a high NA (UPLSApo60XW, NA = 1.2, Olympus) collected the transmitted sample field, and a tube lens 2 and an additional 4-f array magnified the image on a global-shutter camera (DEV-ORX-71S7M, Teledyne FLIR). For off-axis holography, a reference plane wave was combined with the sample field using a second 90:10 beam splitter and linear polarizer. To reconstruct a tomogram, 91 holograms were recorded at different angles of incidence for the plane waves.

For correlative 3D fluorescence imaging, we used a polarizing beam splitter to separate the scanning plane waves for ODT and structured illumination patterns for 3D-SIM into transmitted and epi-illumination modes, respectively. For the validation experiments in Fig. 2 and NIH3T3 cells, we implemented widefield fluorescence microscopy. To image the fiducial markers in Fig. 1 and Fig. 3 with higher spatial resolution, we implemented 3D-SIM, a high-resolution fluorescence microscopy[18,21,45]. Raw 3D-SIM image stacks were acquired with five pattern phases spaced $2\pi/5$, three pattern orientations spaced 60° apart, and the axial translation of an objective lens equipped with a piezoelectric Z stage (P-721. CDQ, Physik Instrumente), at an interval of 180 nm. The 3D-SIM images were then reconstructed using a Richardson–Lucy deconvolution algorithm[56,57]. The hardware was controlled using a custom-built LabVIEW code (LabVIEW 2015 SP1 64 bit; National Instruments). The acquisition times were typically 1.5 and 30 s and >10 min for ODT, 3D widefield fluorescence microscopy, and 3D-SIM, respectively. The schematic diagram for the optical setup is described in the Supplementary Fig. 1.

**Cell preparation**. MDCK cells were cultured in Dulbecco's Modified-Eagle Medium (DMEM, Gibco, CA, USA) supplemented with 10% (v/v) bovine serum (Gibco) at 37 °C within a humidified incubator with 5% $CO_2$. Live NIH-3T3 cells were cultured in DMEM with 10% (v/v) bovine serum at 37 °C in a humidified incubator with 10% $CO_2$. Cells were confirmed to be mycoplasma-free using a BioMycoX Mycoplasma PCR Detection Kit (Cellsafe, Republic of Korea). Cells were transfected with paxillin-pEGFP (a gift from Rick Horwitz; Addgene plasmid #15233) using a microporator (Neon Transfection System, Invitrogen, MA, USA) according to the manufacturer's instructions. Transfected cells were recovered overnight in an incubator and replated on the hydrogels under each condition.

Human CD8$^+$ T cells were isolated from the peripheral blood of a healthy donor. Whole blood samples were kindly donated by healthy donors within the KAIST. All procedures were approved by the Internal Review Board (IRB) of KAIST (approval no. KH2017-004) and followed the Helsinki Declaration of 2000. Informed consent was obtained from all participants prior to blood collection. Peripheral mononuclear cells were collected by density gradient centrifugation and enriched using a CD8$^+$ T cell Isolation Kit (Miltenyi Biotec, Bergisch Gladbach, Germany) according to the manufacturer's protocol. Isolated T cells were then activated and expanded using magnetic microspheres coated with anti-CD3/CD28 antibodies (Dynabeads; Thermo Fisher,

MA, USA) according to the manufacturer's protocol for 72 h. During this period, the resting T cells were prepared in a complete medium only. The collected CD8[+] T cells were resuspended in prewarmed RPMI-1640 (Gibco, CA, USA) supplemented with 10% (v/v) fetal bovine serum and 1% (v/v) penicillin/streptomycin (Sigma-Aldrich, MO, USA) and aliquoted before imaging.

During imaging, all the cells were mounted on a commercial Petri dish compatible with our ODT setup (Tomodish; Tomocube Inc, Republic of Korea) and maintained in a custom incubation stage with 5 mL of corresponding complete medium. During the experiment of drug treatment on NIH-3T3 fibroblasts, the cytochalasin D solution was prepared by mixing with 1 mL complete medium and added to the dish.

**Substrate fabrication**. Polyacrylamide (PA) gel was fabricated according to established protocols[4,58]. The stiffness of PA-gel was adjusted by altering the ratio of acrylamide to bis-acrylamide[59] and polymerized with 0.15% TEMED (Sigma-Aldrich, MO, USA). For a stable bond between the glass and PA-gel, the surface of the bottom glass was coated with silane prior to gel fabrication. During polymerization, we mixed fluorescent beads (diameter 200 nm; FluoSpheres; Life Technologies, CA, USA) with the gel and sandwiched the PA gel with a cover glass of 12 mm diameter and bottom glass to approximately 30 μm of height. For viability to cells, the polymerized PA-gel was treated with 1 mg/mL sulfosuccinimidyl-6-(4-azido-2-nitrophenylamino) hexanoate (Sulfo-SANPAH; Proteochem, UT, USA) in 50 mM HEPES buffer (Life Technologies) and 50 gm/mL collagen type I (Pure-Col; Advanced BioMatrix, CA, USA). We additionally supplemented 16 μg of streptavidin-conjugated acrylamide (Invitrogen[TM], MA, USA) for antibody immobilization per 1 mL of PA-gel mixture for culturing T cells[60]. The concentration of beads was set at three beads per 1 μm[3], which was heuristically optimized to compromise image fidelity and sampling for 3D particle tracking. The gel substrate for adherent cells had a Young's modulus of 11 kPa. The gel substrate for T cells had Young's modulus of 1.2 kPa and was coated with 5 μg/mL each of anti-CD3 (OKT3; 317326; BioLegend, CA, USA) and anti-CD28 (CD28.6; # 16-0288-85; Invitrogen) antibodies according to a previously reported protocol[6]. The timing and concentrations are

**RI reconstruction algorithm**. From raw off-axis holograms, we retrieved the 2D amplitude and phase images using a phase-retrieval algorithm followed by a phase-unwrapping process[61]. From the retrieved 2D field images, we obtained an initial RI estimate using the Fourier diffraction theorem based on the Rytov approximation[62]. To reduce the artifacts caused by noise and the missing cone problem, we implemented a custom algorithm for a fast-iterative shrinkage/thresholding algorithm with constrained total variation minimization[15] (TV-FISTA), which set out to minimize the following loss function:

$$\arg\min_{\mathbf{x}} \frac{1}{2} \|\mathbf{y} - \mathbf{A}\mathbf{x}\|_2^2 + \tau \|\nabla \mathbf{x}\|_1 \qquad (1)$$

where **y** is the input RI image, **x** is the recovered image, **A** is the convolutional operator defined by the point spread function of the optical system, and τ is the regularization parameter for the $l_1$ TV term, $\|\nabla \mathbf{x}\|_1$. To accelerate the deconvolution process, we initially down-sampled the image three times, deconvolved the data (step size $\alpha = 0.003$, $\tau = 0.003$, inner iteration = 100, outer iteration = 75), up-sampled it to the original size, and deconvolved it to the final output image (step size $\alpha = 0.006$, $\tau = 0.006$, inner iteration = 50, outer iteration = 25).

**Estimation of 3D displacement distribution**. To selectively analyze the 3D displacement of beads, we applied a 3D Gaussian-based high-pass filter[63] such that

$$K(i, j, k) = \frac{1}{K_0}\left[\frac{1}{B}\exp\left(-\frac{i^2 + j^2 + k^2}{4}\right) - \frac{1}{w_{xy}^2 w_z}\right] \qquad (2)$$

, with normalization constants, $B = [\sum_{i=-w}^{w} \exp(-i^2/4)]^3$, $K_0 = 1/B[\sum_{i=-w}^{w} \exp(-i^2/2)]^2 - B/w_{xy}^2 w_z$, where $w_{xy}$ and $w_z$ are the lateral and axial window sizes defined as 500 nm and 1 μm, respectively. We then axially removed the cellular region by manually delineating a rectangular mask. To ensure that the embedded beads were locally displaced, we globally registered the processed tomograms using a Fourier transform-based subpixel image-correlation algorithm[19]. The typical subset volume and grid spacing were set as $64^3$ voxels ($11.73^3$ μm$^3$) and 11 voxels (2 μm), respectively.

**Estimation of multidimensional traction force**. To convert the displacement vectors to the multidimensional traction force generated by the cell, $\mathbf{T}(x, y, z = 0)$, we implemented a regularization algorithm with constraints on total variation. Similar to our RI reconstruction algorithm, the algorithm aims to find the traction field **T** that minimizes the following error function:

$$\arg\min_{\mathbf{T}} \frac{1}{2}\|\mathbf{u}_\alpha - \mathbf{G}_{\alpha\beta}\mathbf{T}_\beta\|_2^2 + \tau\|\nabla\mathbf{T}\|_1 \qquad (3)$$

where **u** and $\tau$ denote a $3 \times 1$ displacement vector and regularization parameter, respectively. $G_{\alpha\beta}$ is a $3 \times 3$ dyadic Green's function for a linear elastic medium occupying a half-infinite plane[4,64] such that

$$G_{\alpha\beta}(\mathbf{r}) = \frac{1+\nu}{2\pi E}\begin{bmatrix} \frac{2(1-\nu)\rho+z}{r(r+\rho)} + \frac{x^2\{2\rho(\nu\rho+z)+z^2\}}{\rho^3(\rho+z)^2} & \frac{xy\{2\rho(\nu\rho+z)+z^2\}}{\rho^3(\rho+z)^2} & \frac{xz}{\rho^3} - \frac{(1-2\nu)x}{\rho(\rho+z)} \\ \frac{xy\{2\rho(\nu\rho+z)+z^2\}}{\rho^3(\rho+z)^2} & \frac{2(1-\nu)\rho+z}{r(r+\rho)} + \frac{y^2\{2\rho(\nu\rho+z)+z^2\}}{\rho^3(\rho+z)^2} & \frac{yz}{\rho^3} - \frac{(1-2\nu)y}{\rho(\rho+z)} \\ \left\{\frac{1-2\nu}{\rho(\rho+z)} + \frac{z}{\rho^3}\right\}x & \left\{\frac{1-2\nu}{\rho(\rho+z)} + \frac{z}{\rho^3}\right\}y & \frac{z^2}{\rho^3} + \frac{2(1-\nu)}{\rho} \end{bmatrix} \qquad (4)$$

detailed in Supplementary Fig. 2 and Supplementary Note 1.

where $\rho = (x^2 + y^2)^{1/2}$, $E$ is Young's modulus, and $\nu$ is Poisson's ratio, which is approximately estimated to be 0.5. Each traction vector on a gel position was fitted from $10 \times 10 \times 2$ neighboring displacement vectors. The parameters were typically set at 200 inner iterations and 240 outer iterations, 0.01 for the step size $\alpha$, and 0.02 for the regularization parameter, $\tau$, respectively, and appropriately adjusted. To prevent overregularization at the expense of additional error estimation, we modified TV-FISTA to a monotone version[15].

**Quantitative analysis**. We approximated protein densities from the reconstructed RI values using RI increment per protein concentration[25,65], $\alpha = dRI/dc = 0.185$ mL/g. By subtracting the buoyancy of the medium, we approximated the pressure by cell

weight as the gravitational pressure owing to the dry mass. Gravitational acceleration was set at $9.8 \, m/s^2$. Images were represented and analyzed using MATLAB R2022b (MathWorks Inc.).

**Statistics and reproducibility**. All experiments were performed at least twice or in a time-lapse manner (Fig. 4) to verify reproducibility. All data points represent readings taken from each spatial grid of microscopy images. For statistical hypothesis testing in Supplementary Fig. 5, we determined p values using the two-sided, unpaired Student's *t*-test.

**Reporting summary**. Further information on research design is available in the Nature Portfolio Reporting Summary linked to this article.

## Data availability

The representative datasets used in Fig.1 are available at https://github.com/moosunglee/RI_TFM.

## Code availability

The representative codes for computing the displacement vectors and traction forces are available at https://github.com/moosunglee/RI_TFM.

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

## Acknowledgements
The code for multidimensional traction force was inspired by the TFM algorithm in J.J. Fredberg's lab at the Harvard T.H. Chan School of Public Health. This work was supported by National Research Foundation of Korea (2015R1A3A2066550, 2022M3H4A1A02074314, NRF-2020M3A9E4039658), KAIST Institute of Technology Value Creation, Industry Liaison Centre (G-CORE Project) grant funded by the Ministry of Science and ICT (N11230131) and Institute of Information & communications Technology Planning & Evaluation (IITP; 2021-0-00745) grant funded by the Korea government (MSIT2018K000396).

## Author contributions
M.L., H.J. and C.L. performed and analyzed the experiments. M.L. and H.J. provided analytical tools. B.R.D. and M.J.L. supported the experiments. J.S., W.H. and Y.P. supervised the project. All authors wrote the paper.

## Competing interests
The authors declare no competing interests.
