## [Peer Review File · Communications Biology]

Reviewers' comments:

Reviewer #1 (Remarks to the Author):

The manuscript "High resolution assessment of multidimensional cellular mechanics using label-free refractive-index traction force microscopy" by Lee et al presents a new 3D traction force microscopy technique relying on refractive index differences between the sample, the beads and the matrix. This allows not only the acquisition of traction forces but also changes in refractive index of the cells and cellular compartments in a low phototoxic and label-free experimental setting. While this technique is novel and useful for the field, the individual results presented seem somewhat preliminary and a publication in its current form not recommended.

1. A major concern of the study is the low number of repeats shown in the study. To support the claim that the new method is equal and better than FL TFM higher number of repeats are necessary especially to judge the root-mean square errors between the methods as currently given in line 99 of the manuscript.
2. The methods as currently presented are not very clear. To enable reproducibility of the results, the authors should give a more detailed account of the experimental procedure including the gel fabrication, gel stiffness measurement as well as chemical detaching with incubation timings and concentrations. A schematic of the instrument should also be included.
3. The work starts by comparing established fluorescence based traction force imaging with the refractive index (RI) TFM. As shown in Figure 1d) the displacement vectors in the fluorescence and refractive index acquired images do not match. The authors argue that this is first, due to point-based artefacts in the SIM images acquired so they later switch to widefield fluorescence and second, due to the noise through long-time acquisition. The authors should repeat the comparison of RI TFM with widefield FL images as well as provide evidence for the differences in refractive index between the fiducials and the cells. In Figure 1b) the data seems to show that the refractive index of the cell is overlapping with the fiducial markers they use to perform traction force. The authors should show data supporting the differences in the refraction index for example by providing a refractive index measurement of the fiducial makers. Moreover it is not clear why manually removing the cell is necessary for the analysis (line 92, 365). Could the authors segment beads and cells via refractive indices instead?
4. In Figure 2 the authors argue that the addition of spherical glass beads does not alter the performance of their method. If the authors could provide a control where fixed cells are added to the gels this would be a more meaningful control than the current glass spheres, this would support the claim that the refractive index of cells does not alter their measurement. Furthermore, can the authors comment on why in Figure 2c) the addition of the spherical silica beads alters the accuracy in z. More repeats to gather the variation between measurements would be helpful here.
5. In Figure 5 the authors present results showing forces during activation of primary CD8 T cells. The authors should show some marker of T cell activation, such as recording the calcium release as a marker for T cell activation as a pretext for the formation of an immunological synapse, this would be helpful to support this claim. Moreover, more measurements of T cells are needed to claim any observed difference in exerted forces between activated and non-activated cells.

Minor points:

- Y-Axis label for Fig 2c
- Number of repeats should be clearly stated in the captions of each figure
- Code availability statement should be included
- Line 138 "...traction force per unit area.."

Reviewer #2 (Remarks to the Author):

The paper "High-resolution assessment of multidimensional cellular mechanics using label-free refractive-index traction force microscopy" reported the performance and application of RI-TFM. This work would be greatly enhanced if author compares the performance with other kinds of TFM.

Major points :

1. Author claims that "The axial precision of RI-TFM, however, outperformed widefield FL microscopy by 3.6 times." But widefield FL microscopy is already a very traditional imaging technology and is not a suitable comparison object. The author needs to compare RI-TFM with new TFM, such as 3D-SIM-TFM(Nano Lett. 2019, 19, 4427–4434) and aTFM(NATURE COMMUNICATIONS, 2021, 12:2168) etc.
2. How serious is the influence of mechanical noise on 3D-SIM? Author gives up the advantage of 3D-SIM super-resolution ability to reconstruction the displacement map and traction fields.
3. It's maybe better to add light path diagram in supplementary material and describe how it works.
4. The algorithm of estimation traction stress is different from methods of solving the force field in reverse. What makes this difference? How your algorithm works?

Reviewer #3 (Remarks to the Author):

Park and co-workers present in this manuscript a method for measuring cellular 3D traction forces in a planar setup with a combination of optical refractive index tomography and computational image analysis. Moreover, they combine this method with structured illumination for imaging cellular features at high resolution.

This is an outstanding manuscript. The methodology is impressive and well-explained. The limitations and strengths of the proposed approach are carefully assessed and clearly described. The method itself, although relying on quite demanding hardware, holds exciting potential for future research. I have read this manuscript carefully and could not find any flaw. Publication is recommended.

Comments to the Authors and Response

Reviewer #1

The manuscript “High resolution assessment of multidimensional cellular mechanics using label-free refractive-index traction force microscopy” by Lee et al presents a new 3D traction force microscopy technique relying on refractive index differences between the sample, the beads and the matrix. This allows not only the acquisition of traction forces but also changes in refractive index of the cells and cellular compartments in a low phototoxic and label-free experimental setting. While this technique is novel and useful for the field, the individual results presented seem somewhat preliminary and a publication in its current form not recommended.

We very much appreciate the reviewer’s scrupulous attention to our work. Following the reviewer’s constructive suggestions, we have revised the manuscript and strengthened its quality. We expect that the amended presentation of the results addresses the essential points.

[1] A major concern of the study is the low number of repeats shown in the study. To support the claim that the new method is equal and better than FL TFM higher number of repeats are necessary especially to judge the root-mean square errors between the methods as currently given in line 99 of the manuscript. We agree with the reviewer that our method should demonstrate superior repeatable performance. Following the reviewer's suggestion, we have additionally compared four sets of three-dimensional (3D) displacement vectors utilized in Fig. 3 (RFig. 1). The results show that fluorescence-based traction force microscopy (FL-TFM) exhibited pointwise artefacts, even where the cells were not present. These results validated more robust performance of our method, aligning with the results shown in Fig. 1.

As explained in the manuscript, the reconstruction artefacts in FL-TFM stem mainly from the long acquisition time of structured illumination microscopy (SIM), which exceeded 10 minutes and made the technique susceptible to mechanical noise and motion artefacts. In addition, the significant photobleaching and the vignetting problems restricted the consistent reconstruction performance of SIM across a wide FOV, despite a proper post-correction process [GigaScience, 9(4), giaa035, 2020]. We have included these data in Supplementary Figure 4 and discussed them in the revised manuscript.

RFigure 1. (a, b) XZ and XY cross-sections of refractive index (RI; grey) and FL (paxillin-EGFP; green) and 3D displacement vector for NIH3T3 (a) cell 1 and (b) cell 2, before and 1 h after treatment with cytochalasin D (5 μM). Std indicates the standard deviations of the displacement vectors along the x, y, and z directions.

[2] The methods as currently presented are not very clear. To enable reproducibility of the results, the authors should give a more detailed account of the experimental procedure including the gel fabrication, gel stiffness measurement as well as chemical detaching with incubation timings and concentrations. A

schematic of the instrument should also be included.

We appreciate your feedback regarding the lack of clarity in the gel preparation instructions. We have addressed this by providing a detailed procedure with a graphical illustration (RFig. 2). We would also like to clarify that we followed the protocol and used the estimated gel stiffness value based on the previous study [Current Protocols in Cell Biology 47, 10.16.11-10.16.16 (2010)]. This can also be found in our Supplementary Materials.

RFigure 2. Schematic illustration of the gel fabrication. (a, b) Procedures for coating (a) extracellular matrix (ECM) for NIH-3T3 and MDCK cells and (b) antibodies for T cells, respectively.

[3] The work starts by comparing established fluorescence based traction force imaging with the refractive index (RI) TFM. As shown in Figure 1d) the displacement vectors in the fluorescence and refractive index acquired images do not match. The authors argue that this is first, due to point-based artefacts in the SIM images acquired so they later switch to widefield fluorescence and second, due to the noise through long-time acquisition. The authors should repeat the comparison of RI TFM with widefield FL images as well as provide evidence for the differences in refractive index between the fiducials and the cells. In Figure 1b) the data seems to show that the refractive index of the cell is overlapping with the fiducial markers they use to perform traction force. The authors should show data supporting the differences in the refraction index for example by providing a refractive index measurement of the fiducial makers. Moreover it is not clear why manually removing the cell is necessary for the analysis (line 92, 365). Could the authors segment beads and cells via refractive indices instead?

Thank you for your insightful comments. The reviewer's suggestions can be categorized as follows: (1) comparing our method with widefield FL-TFM, (2) comparing refractive indices (RIs) between the cell and fiducial markers, and (3) applying sophisticated segmentation techniques to our sample. We have addressed the suggestions as below.

We first compared our method with widefield FL-TFM in Fig. 1 (RFig. 3). Although the displacement vectors from the three different imaging methods exhibit similar patterns, the data from

the widefield image resulted in the most significant deviation across the edge of the field of view. We quantified this deviation by estimating the root-mean-square errors (RMSEs) between SIM-based TFM and widefield TFM. The result indicated the RMSEs of 5.75, 34.43, and 31.02 nm in the x , y , and z directions respectively. Compared to RI-TFM, the lower precision in widefield FL-TFM is attributed to its inferior spatial resolution, as consistent with Fig. 1 and RFig. 1. These results support the superior performance of RI-TFM over FL-based methods. We have also discussed this accordingly in the main text to improve its clarity and flow.

RFigure 3. Displacement vectors computed from RI (cyan; left), 3D FL-SIM (magenta; right), and 3D widefield FL (yellow; right) images. RMSE, root mean square error between the vectors from FL-SIM and those from widefield FL along the x , y , and z directions.

Next, we explain how our RI tomogram could distinguish the fiducial markers from the cell membrane near the substrate. First, we chemically embedded the fiducial markers in the gel substrate, in order to exclude the possibility that the cell membrane assimilated the fiducial markers. This enabled us to spatially segregate the axial locations of fiducial markers from the region of attached cells. Moreover, we quantitatively confirmed that the RI value of the fiducial markers were significantly higher than the cell membrane. To demonstrate this, we used correlative images of RI tomography and FL-SIM to image the fiducial markers at the substrate boundary (RFig. 4a). The images revealed that the fiducial markers exhibited 0.02 higher RI values than the cell membrane (RFIGs. 4b-c).

Please note that both imaging modalities provided a full-width half-maximum of the 385.4 nm laterally and 1093.9 nm axially for RI-TFM, and 512.8 nm laterally, and 791.5 nm axially for FL-SIM. These numbers are double the actual system resolution we determined in our previous research [eLife 9:e49023 (2020)], because (1) we used beads with a diameter of 200 nm, which were already larger than the lateral resolution, and (2) we adjusted the pixel size slightly larger than the actual resolution (183.4 nm) to speed up the computation. Nevertheless, the obtained results provided sufficient resolution to separate the fiducial markers from the cell membrane with high-pass filtering and manually removing the cellular region. We have included the detailed presentation in the Supplementary Figure 3 and the Discussion of the revised manuscript.

RFigure 4. Distinctness of fiducial markers in RI-TFM. (a) 2D images of RI and FL SIM on the substrate surface. (b-c) Magnified image of a single bead. (b) XY cross-sections, line plots, and corresponding lateral full-width half-maximum. (c) XZ cross-sections, line plots, and corresponding axial full-width half-maximum.

Regarding the segmentation, we could improve our analysis by employing a sophisticated technique, such as deep-learning-based methods, as demonstrated in our previous work [eLife 9:e49023 (2020); Biomedical Optics Express, 14(9), 4567-4578 (2023)]. However, this would necessitate extensive data curation and training, which is beyond the scope of the present work. We respectfully suggest that an independent follow-up study should test this idea.

[4] In Figure 2 the authors argue that the addition of spherical glass beads does not alter the performance of their method. If the authors could provide a control where fixed cells are added to the gels this would be a more meaningful control than the current glass spheres, this would support the claim that the refractive index of cells does not alter their measurement. Furthermore, can the authors comment on why in Figure 2c) the addition of the spherical silica beads alters the accuracy in z. More repeats to gather the variation between measurements would be helpful here.

The referee raised an important point on the validation experiments, which we had also considered. The analysis shown in Fig. 1d and RFig. 1 demonstrates the robust performance of our method in the presence of live cells. We believe that these additional data would be sufficient to support the claim in Fig. 2.

As the reviewer correctly points out here, our concern was whether the presence of samples on the substrate might alter the performance of our method. As shown in Fig. 2c, we found that the addition of samples did indeed alter the accuracy in z. However, this effect was not significant, but still provided better accuracy than FL-TFM.

We would also like to clarify that the data presented in Fig. 2c is a collective histogram of six measurements over the different displacements of the stage, and each measurement contained more than 2,700 data points. We have also presented a supplementary time-lapse video 1 showing our robust measurement performance, for which we believe that these measurements provided sufficient data to validate our arguments.

[5] In Figure 5 the authors present results showing forces during activation of primary CD8 T cells. The authors should show some marker of T cell activation, such as recording the calcium release as a marker for T cell activation as a pretext for the formation of an immunological synapse, this would be helpful to support this claim. Moreover, more measurements of T cells are needed to claim any observed difference in exerted forces between activated and non-activated cells.

These are insightful ideas that we had not considered. Firstly, the use of calcium imaging would be beneficial to our work as it would provide additional information on the dynamics of T cell signalling. However, this application is beyond the scope of our study, in which we were mainly interested in the effect of the activated T cell status on the traction force. For this purpose, we implemented a commercially established CD8+ T cell Isolation Kit (Miltenyi Biotec, Bergisch Gladbach, Germany) according to the manufacturer's protocol. We respectfully believe that correlative calcium imaging of live human T cells should be tested in an independent follow-up study.

However, we recognize the need to repeat the experiments to confirm our findings. We have therefore performed an experiment to statistically analyse the traction force and the surface protein density (RFig. 5). We plotted 6 and 10 data from the resting and the activated T cells, respectively (RFig. 5a). The statistics showed that the activated T cells exhibited the significantly higher normal and shear traction forces. Notably, the median value of the surface protein density was not significantly different, whereas its maximum value was significantly different between the resting and activated T cells (RFig. 5b). This is consistent with our observation that the activated T cell exhibited high-density protein clusters (Fig. 5d). Taken together, our statistical analysis confirmed our previous observation in Fig. 5. We have included this in Figure 5—Supplementary Figure 5.

RFigure 5. Statistical analysis of traction forces and surface protein density of human CD8+ T cells. (a) Representative RI cross-sectional images of two resting (top) and two activated (bottom) T cells. (b) Quantification of median (top) and maximum (bottom) values for normal traction (left), shear traction (middle), and surface protein density (right). $N = 6$ for resting, and 10 for activated T cells. Mean \pm standard deviation is shown for each boxplot. Each boxplot indicates the median, upper, lower quartiles,

and 1.5x interquartile range of each population. *: $P < 0.05$, **: $P < 0.01$, ***: $P < 0.001$ for the two-sided, unpaired Student's t-test.

[6] Minor points:

- Y-Axis label for Fig 2c
- Number of repeats should be clearly stated in the captions of each figure
- Code availability statement should be included
- Line 138 "...traction force per unit area.."

Thank you for noticing the typos, which we have corrected in the revised manuscript. In addition, the representative dataset and codes are available on Github (https://github.com/moosunglee/RI_TFM), which will promote useful applications of our method.

Reviewer #2 (Comments to the Author):

The paper “High-resolution assessment of multidimensional cellular mechanics using label-free refractive-index traction force microscopy” reported the performance and application of RI-TFM. This work would be greatly enhanced if author compares the performance with other kinds of TFM.

We thank the reviewer for the constructive comments. The reviewer’s suggestions were very helpful in drawing our attention to the lack of the comparison with other relevant references. We have revised the manuscript accordingly, improving the clarity and flow of information.

[1] Author claims that “The axial precision of RI-TFM, however, outperformed widefield FL microscopy by 3.6 times.” But widefield FL microscopy is already a very traditional imaging technology and is not a suitable comparison object. The author needs to compare RI-TFM with new TFM, such as 3D-SIM-TFM(Nano Lett. 2019, 19, 4427–4434) and aTFM(NATURE COMMUNICATIONS, 2021, 12:2168) etc.

We thank the reviewer for your insightful suggestions. In the revised Discussion section, we have included the recommended references and compared them with our work.

[2] How serious is the influence of mechanical noise on 3D-SIM? Author gives up the advantage of 3D-SIM super-resolution ability to reconstruction the displacement map and traction fields.

We thank the reviewer’s comments for correctly identifying this issue. We agree with the reviewer that our method should demonstrate superior repeatable performance. Following the reviewer's suggestion, we have additionally compared four sets of three-dimensional (3D) displacement vectors utilized in Fig. 3 (RFig. 1). The results show that fluorescence-based traction force microscopy (FL-TFM) exhibited pointwise artefacts, even where the cells were not present. These results validated more robust performance of our method, aligning with the results shown in Fig. 1.

As explained in the manuscript, the reconstruction artefacts in FL-TFM stem mainly from the long acquisition time of structured illumination microscopy (SIM), which exceeded 10 minutes and made the technique susceptible to mechanical noise and motion artefacts. In addition, the significant photobleaching and the vignetting problems restricted the consistent reconstruction performance of SIM across a wide FOV, despite a proper post-correction process [GigaScience, 9(4), g1aa035, 2020]. We have included these data in Supplementary Figure 4 and discussed them in the revised manuscript.

RFigure 1. (a, b) XZ and XY cross-sections of refractive index (RI; grey) and FL (paxillin-EGFP; green) and 3D displacement vector for NIH3T3 (a) cell 1 and (b) cell 2, before and 1 h after treatment with cytochalasin D (5 μ M). Std indicates the standard deviations of the displacement vectors along the x , y , and z directions.

Moreover, 3D-SIM takes 30 times longer than conventional FL methods, which adversely affects our ability to access rapid, long-term live cell dynamics.

[3] It's maybe better to add light path diagram in supplementary material and describe how it works. Thank you for the constructive comment. We have added the diagram for the optical setup in Supplementary Figure 1 (RFig. 6).

RFigure 6. Schematic diagram of the optical setup. DMD: digital micromirror device; PBS: polarizing beam splitter; BS: beam splitter. Green rays indicate the beam path for optical diffraction tomography (ODT). Yellow rays indicate the beam path for fluorescence excitation (FL excitation). Red rays indicate the beam path for emitted fluorescence light (FL emission).

[4] The algorithm of estimation traction stress is different from methods of solving the force field in reverse. What makes this difference? How your algorithm works?

Thank you for bringing this to our attention. Before proceeding with our explanation, we would like to clarify that there were typos in the “Estimation of multidimensional traction stress” section of the Method section. Please note that we calculated the traction force field, not the traction stress, from the 3D displacement vectors throughout the manuscript.

In response to the reviewer's comments, we would like to provide more details about our method for solving the force field using the gel's displacement vector. The problem of solving the displacement vector of the gel, $\mathbf{u}(\mathbf{r})$, caused by the cell's traction force, $\mathbf{T}(\mathbf{r})$, is expressed by the following linear integral equation,

$$\mathbf{u}_\alpha(\mathbf{r}) = \int \sum_{\beta=1}^3 \mathbf{G}_{\alpha\beta}(\mathbf{r} - \mathbf{r}') \mathbf{T}_\beta(\mathbf{r}') d\mathbf{r}' \quad (1)$$

, where $\mathbf{G}_{\alpha\beta}$ is a 3×3 dyadic Green's function for a linear elastic medium occupying a half-infinite plane. Its inverse problem, however, is an ill-posed problem due to the limited measurement range of $\mathbf{u}(\mathbf{r})$. To resolve this issue, we implemented an iterative regularization algorithm by imposing constraints on total variation. The relevant reference can be found [Biophysical J., 120, 3079-3090 (2021)] and included in the revised manuscript. In addition, the computation algorithm is accessible on Github, and we anticipate that this will provide further information about how the algorithm operates.

Reviewer #3 (Comments to the Author):

Park and co-workers present in this manuscript a method for measuring cellular 3D traction forces in a planar setup with a combination of optical refractive index tomography and computational image analysis. Moreover, they combine this method with structured illumination for imaging cellular features at high resolution.

This is an outstanding manuscript. The methodology is impressive and well-explained. The limitations and strengths of the proposed approach are carefully assessed and clearly described. The method itself, although relying on quite demanding hardware, holds exciting potential for future research. I have read this manuscript carefully and could not find any flaw. Publication is recommended.

We thank the reviewer for the positive review. We have further revised the manuscript according to the comments by the other reviewers, which we believe have significantly strengthened the quality of our research.

REVIEWERS' COMMENTS:

Reviewer #1 (Remarks to the Author):

I thank the authors for addressing the points made in my comments, especially in including Supplementary Figure 5, demonstrating the possibility to use this methodology to gain valuable insights into cell signalling.

Regarding my previous point about RI values, in the new SI Figure 3, if a histogram of RI values for the fiducials and the cell membrane could be included it would be highly appreciated, as it would make the point that, as the authors say, " the RI value of the fiducial markers were significantly higher than the cell membrane" and then go on to quote the value of 0.02, which is currently not accessible from the figure.

The manuscript has made progress and I would recommend it for publication if the point is addressed.

Reviewer #2 (Remarks to the Author):

I have read the revised manuscript. Most of my concerns have been addressed. I recommend this paper to be published.

In addition, I have a small suggestion. I agree the long acquisition time of 3D-SIM may introduce mechanical noise and motion artefacts. But the author needs to be reminded that the latest 3D-SIM imaging usually takes a few seconds, not the 10 minutes mentioned by the author, and the artefacts should be very weak.

Comments to the Authors and Response

Reviewer #1

I thank the authors for addressing the points made in my comments, especially in including Supplementary Figure 5, demonstrating the possibility to use this methodology to gain valuable insights into cell signalling.

Regarding my previous point about RI values, in the new SI Figure 3, if a histogram of RI values for the fiducials and the cell membrane could be included it would be highly appreciated, as it would make the point that, as the authors say, "the RI value of the fiducial markers were significantly higher than the cell membrane" and then go on to quote the value of 0.02, which is currently not accessible from the figure.

The manuscript has made progress and I would recommend it for publication if the point is addressed. We thank the reviewer's positive consideration of our work. Although we considered plotting the histogram, the result showed the higher mean RI value in the cellular region (RFig. 1). However, this is natural as the fiducial markers constitute small volumetric fractions with higher peak RI values than the broad cell membranes. Since the histogram may lead to a misleading interpretation, we decided to present only the linear plots of the fiducial markers in Supplementary Fig. 3.

RFigure 1. Histogram plot of the 1.1- μm -thick (6 voxels) cellular region (cell; blue) and fiducial markers (beads; red) in the vicinity of the gel substrate. The horizontal bar graph indicates mean \pm standard deviation values for each region.

Reviewer #2 (Comments to the Author):

I have read the revised manuscript. Most of my concerns have been addressed. I recommend this paper to be published.

In addition, I have a small suggestion. I agree the long acquisition time of 3D-SIM may introduce mechanical noise and motion artefacts. But the author needs to be reminded that the latest 3D-SIM imaging usually takes a few seconds, not the 10 minutes mentioned by the author, and the artefacts should be very weak.

We thank the reviewer for the constructive and positive review. As correctly pointed out, we have added the relevant discussion in the revised manuscript.